# Evaluation of the engineering properties of asphaltic concrete composite produced from recycled asphalt pavement and polyethylene plastic

David Olukanni[1], Benjamin Oyegbile[1,2]*, Akanimo Ukpeh[1]

1 Department of Civil Engineering, College of Engineering, Covenant University, Ota, Nigeria, 2 Department of Process Engineering, Faculty of Engineering, Stellenbosch University, Stellenbosch, South Africa

* hollander196@yahoo.com

**Data Availability Statement:** The processed data and the images supporting this work has been deposited with figshare. This is the standard practice for data sharing in our field. these files are

## Abstract

This study investigated the suitability of recycled asphalt pavement and polyethylene wastes as coarse aggregate in asphaltic concrete by evaluating the impact of the use of polyethylene polymer wastes and recycled asphalt pavement composite as aggregates on the physical and mechanical properties of the asphaltic concrete. The physical characteristics of the aggregate and bitumen were determined using relevant parametric tests. Recycled asphalt pavement was used to make asphaltic concrete samples using LDPE at 5%, 10%, 15%, RAP at 5% and HDPE at 5%, 10%, 15%, and a mixture of LDPE + HDPE at 5+5%, 7.5 +7.5 and 10+10% RAP at 5% as additives. Marshall Stability test was conducted to assess the mechanical strength of the asphaltic concrete, and the results included information on the aggregate's stability, flow, density, voids filled with bitumen, voids filled with air, and voids in mineral aggregate. In addition, the surface and crystal structure of the aggregates was studied by carrying out a microscopic examination with a Scanning Electron Microscope (SEM) and X-Ray diffraction (XRD). The results obtained from this study demonstrated that RAP, HDPE & LDPE are viable conventional aggregate substitute for asphalt concrete production.

## Introduction

Rapid urbanisation and population growth has led to a substantial increase in the amount of waste being generated worldwide. In the absence of an effective management of municipal waste and recycling measures, public health and the environment will be at risk. HDPE and LDPE which constitute one of the biggest classes of plastic wastes produced in Nigeria may be recycled more effectively by using it as raw materials in roads and other infrastructural construction. This has the potential to reduce environmental degradation from plastic waste and promote the concepts of zero waste and circular economy through materials recycling. Pavement failure is another growing trend in Nigeria which ultimately leads to the excavation of old pavement during repair works. High temperatures and high loads have been identified as

available from Figshare database with the URL: https://doi.org/10.6084/m9.figshare.24204516.v1.

**Funding:** The author(s) received no specific funding for this work.

the main causes of early failure in flexible pavements. There has been a lot of work on developing different strategies for recycling of the excavated pavement materials.

A pavement is a wear-resistant rigid or flexible surface that serves a wide range of purposes such as in facilitating pedestrian and vehicular movement, as parking or public space, and in landing, take-off and taxing of aircrafts at the airport. Cobblestones and granite paving stones were common in the past, but they have been mostly replaced with compacted foundation courses of asphalt or concrete. Flexible pavement is one of the most commonly used form of pavement around the world [1]. Typically, a flexible pavement, also known as asphalt or tarmac, consists of many layers which include: a top layer (surface), a second layer (subbase), and a third layer (subgrade). A flexible pavement's top layer is made out of hot-mix asphalt (HMA). Untreated aggregates are often used in the second layer, although asphalt, foamed bitumen, Portland cement, and other stabilising materials may also be used. As a general rule, the third layer is made out of local aggregate material, and it is generally topped with cement or lime. Flexible pavements can typically last between 20 and 30 years if well-maintained [2].

Several research works on the applications of a wide range of waste materials (e.g., waste plastics, quarry waste, fuel ash, natural fibres, waste glass, waste bricks, waste ceramics, tire rubber, steel slag, recycled asphalt pavement (RAP) etc.) as modifiers in the asphaltic concrete mix to improve wear-resistance and the mechanical properties of the pavement against deformations and breakage have been published [1, 3–7]. Many of these comprehensive reviews provides timely information on the latest developments on the reuse of waste products in road construction. However, one drawback in most of these studies is lack of any meaningful analysis on the long-term environmental impact of these substitute materials in terms of the risk of soil and groundwater pollution as well as econometric analysis of their utilization as construction material.

Khan et al. [8] investigated the rheological properties of a modified bitumen (PG 64–10) in asphalt mix using LDPE, HDPE, and crumb rubber with respect to rutting and fatigue cracking. These properties were evaluated on the basis of parameters such as complex modulus (G), phase angle (δ) from a dynamic shear rheometer measurement. For instance, asphaltic pavements have been shown to be less susceptible to rutting if recycled asphalt pavement (RAP) is used as aggregate substitute in the asphalt mix. Improved elasticity can also be achieved by using recycled asphalt. The results of the investigation showed that an improvement in the elasticity of the modified bitumen binder with less susceptibility to the effect of temperature. Their results also showed that using 10% LDPE in the bitumen binder gives the best value for the rutting perimeter at all measured temperatures and hence, offers the best protection against rutting.

In a similar research work, Hassani et al. [9] explored the use of plastic waste—poly-ethylene terephthalate (PET) as aggregate replacement in asphaltic concrete mix. Industrial scale PET granules used for in the study which has comparable properties to the waste PET plastc. The results from the study show that all the test specimens met the design criteria set by the Iranian Asphalt Institute in terms of their mechanical strength with the only exception being the flow measurement In addition, the stability of asphalt concrete with substituted aggregates decreases compared to the control specimen. This trend which was attributed to increased friction between the PET granules increases as the proportion of the substituted aggregates increases.

Similar communications by Ezemenike et al. and Olukanni et al. [10, 11] reported their findings on the use steel slag and a combination of hydrated slag, glass powder and cement modifiers as aggregate replacements in asphalt concrete production. Stability of the asphalt concrete produced was evaluated on the basis of Marshall stability test and scanning electron microscopy. Both of these studies reported satisfactory performance of the asphaltic concrete test specimens in terms of the evaluation criteria when compared with the minimum Marshall stability value set by the Asphalt institute. The microstructure analysis performed using SEM also showed that the asphaltic concrete specimen with glass powder the least inter-particulate

space. Considering the current global environmental concern and the multitude of waste materials available as a potential replacement option for aggregates in asphalt concrete, it is not surprising to find significant research efforts in this field.

Plastic pollution represent one of the biggest environmental problem worldwide and this poses a serious threat to terrestrial and marine life. The aim of this study is to assess the production of asphalt with recycled asphalt pavement (RAP) and polyethylene polymers (HDPE and LDPE) as aggregate subsitutes. The objectives of this research are as follows: to determine the physical and mechanical properties of the aggregate materials used in this study by carrying out the necessary material tests on the natural bitumen binder and RAP such as penetration test, ductility test, softening point test, viscosity test, specific gravity test, aggregate impact value, aggregate abrasion test, bulk density and void percentage, elongation index test, flakiness test etc.; characterization of the produced asphaltic concrete composite in terms of its surface structure and mechanical properties using Scanning Eletron Microscopy and X-Ray Diffraction (XRD) Techniques.

## Materials and methods

### Sample preparation and characterization

Naturally occurring bitumen of grade 70–100 obtained from lake deposit was used for the asphaltic concrete mix. The bitumen sample was characterized in the laboratory to determine its physical properties while an optimum bitumen content (o.b.c) of 5.1% determined from the Marshall stability test of the control sample was chosen for the design mix for the rest of the test specimen [9]. Tap water with a neutral pH obtained at the Covenant University campus, Ota was used in the asphalt mix preparation and curing. Granites obtained from a local quarry site was used as the natural coarse aggregates in the preparation of the reference asphaltic concrete samples. Stone dust also acquired within the University campus was used as filler in the asphalt production. Recycled pavement made of asphalt was utilized as a partial substitute for the aggregates in the mix. Recycled plastic pipes recovered from discarded plumbing fixtures served as High Density Polyethylene (HDPE) and pulverized nylon bags served as the Low Density Polyethylene (LDPE) plastic wastes respectively. Plastic pipe waste and waste nylon bags were collected, cleaned, and then cut into smaller pieces for use as aggregates substitute alongside (RAP). The HDPE and LDPE were then melted separately and then crushed using a mechanical grinder to convert the waste materials to suitable coarse aggregate sizes (~4.75mm-12.75mm). The characteristic material properties of the natural bitumen binder and the aggregates and are shown in Tables 1 and 2 below.

### Preparation and characterization of asphaltic concrete test samples

Asphaltic concrete test specimens were prepared in triplicates for each design mix formulations and the control mix according to the specification shown in Table 3. The control mix which was prepared from bitumen and natural aggregates without any aggregate substitution served as a reference sample for the study. All samples of the asphaltic concrete were prepared

**Table 1. Characteristic properties of natural bitumen used for the study.**

| Bitumen | Characteristic values |
|---|---|
| Penetration test | 127mm |
| Ductility test | 80cm |
| Softening test | 41˚c |
| Viscosity test | 520secs |
| Specific gravity | 1.02 |

**Table 2. Characteristic properties of coarse aggregate and RAP used for the study.**

| Tests | RAP | Natural Aggregates |
|---|---|---|
| Los Angeles abrasion | 37.5% | 33.2% |
| Aggregate impact value | 35% | 25.2% |
| Specific gravity | 2.60 | 2.65 |
| Water absorption | 0.93% | 0.71% |
| Bulk density | 2235kg/m$^3$ | 2450kg/m$^3$ |
| Elongation index | 28% | 24% |
| flakiness index | 28% | 25% |

and cured under laboratory conditions. Tests were thereafter performed on the asphaltic concrete samples after preparation and curing to evaluate the effect of the substituted waste aggregates—recovered recycled asphalt pavement (RAP) and polyethylene polymers (HDPE and LDPE) on the microstructure and mechanical properties of the asphaltic concrete. Marshall stability test, Scanning Electron Microscopy (SEM) and X-ray diffraction nalysis (XRD) analysis were subsequently carried out on the prepared test samples shown in Fig 1.

## Marshall stability test

Marshall Stability test is a popular and proven method to measure the load bearing and flow rate of asphaltic concrete specimens and it is usually performed using a Marshall Stability test machine. It is also used in carrying out the density-voids analysis, strength, and flexibility evaluation, as well as determining if the provided bituminous mixture is suitable for use as a pavement material and to determine the optimum content. Marshall stability test was performed in this study using a similar procedure adopted by Hassani et al. [9].

## Crystal structure characterization

X-Ray diffraction analysis was performed on the asphalt samples to determine their molecular structure and crystallography. The structural attributes of the asphalt concrete samples are dependent on their distinct microstructural formations. X-Ray diffraction was performed using an X-ray diffractometer.

## Surface structure characterization

The micro and failure analysis of solid inorganic materials may be done successfully using the Scanning Electron Microscopy to obtain a high resolution image. Using a high magnification,

**Table 3. Aspaltic concrete test samples' specification.**

| Sample No. | Recycled Aggregate Design Mix |
|---|---|
| Sample 1 (control) | 0% RAP+0% LDPE |
| Sample 2 | % RAP + 5% LDPE |
| Sample 3 | 5% RAP + 5% HDPE |
| Sample 4 | 5% RAP + 10% LDPE |
| Sample 5 | 5% RAP + 10% HDPE |
| Sample 6 | 5% RAP + 15% LDPE |
| Sample 7 | 5% RAP + 15% HDPE |
| Sample 8 | 5% RAP + 5% LDPE & 5% HDPE |
| Sample 9 | 5% RAP + 7.5% LDPE & 7.5% HDPE |
| Sample 10 | 5% RAP + 10% LDPE & 10% HDPE |

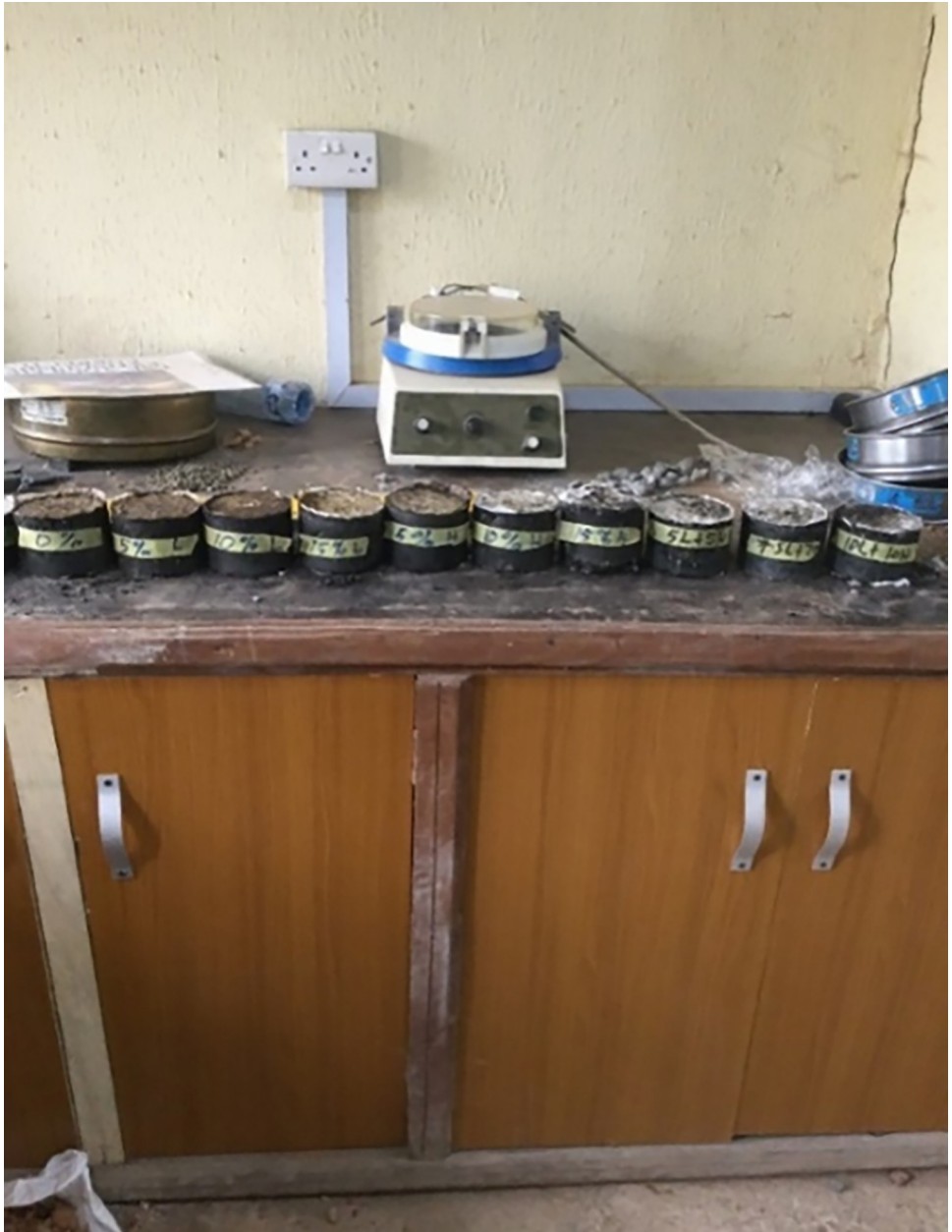

**Fig 1. Test samples of the asphaltic RAP-HDPE-LDPE concrete.**

this test produces high-quality images and exceptionally accurate measurements of extremely small features and objects. Scanning an asphalt sample with an electron beam produces an enlarged picture that may be subjected to a more detailed analysis. SEM is a high-performance technique used to investigate the structure of materials. SEM microscopy has two benefits over optical microscopy (OM) in terms of resolution and magnification, as well as a high field depth The SEM analysis was performed at the central analytical facility at the Covenant University, Ota.

## Results and discussions

The results of the material tests carried out on the bitumen samples, aggregates, recycled asphalt pavement (RAP), and the SEM and microstructural analysis of the asphaltic concrete samples made with and without aggregate replacements are presented in this section. These results show how well the modified asphalt samples compare to a control sample, and the extent to which the concrete samples meet the specified criteria for pavement contruction. The microstructural analysis results present the chemical composition of the prepared asphalt samples which also influence their overall properties.

### Marshall stability test results

The Marshall Stability test was used to assess the asphalt samples' acceptability for use as a road pavement material by measuring the following properties of the asphaltic concrete samples: Specific gravity of mix, volume of bitumen, volume of aggregates, voids in mineral aggregate (VMA), Void filled with bitumen (VFB), Stability and Value of flow. Figs 2–4 and Table 4 show the results of the Marshall stability test performed on the cured asphalt concrete samples.

All the asphalt concrete sample recorded ttest values that are within the Federal Ministry of Works 1997 limit for stability as the state minimum recommended stability is 3.5KN. They also meet the standards for flow value as the recommended flow value range is 2-4mm. It was observed that not all the concrete samples met the required void filled with bitumen range of 65–75, as 5% L, 10% L, 7.5% (L+H) and 10% (L+H) had their voids filled with bitumen below the minimum value of 65. The concrete samples with 7.5% (L+H) and 10% (L+H) plastic aggregates did not meet the requirements for air voids as their values were above the stipulated limit of 5. Sample with aggregates (5% RAP + 15% HDPE) is our optimum asphalt sample with the best stability value which is 8.7KN with all other parameters falling within acceptable tolerances. In terms of the stability results, this mix ratio (5% Rap + 15% HDPE) meets the specified standard for use as a pavement wearing or surface course.

### Crystallography analysis of asphaltic concrete samples using XRD

X-Ray diffraction (XRD) examination was conducted on control sample, optimum sample and the mix of both HDPE and LDPE to determine the crystal structure of the asphaltic

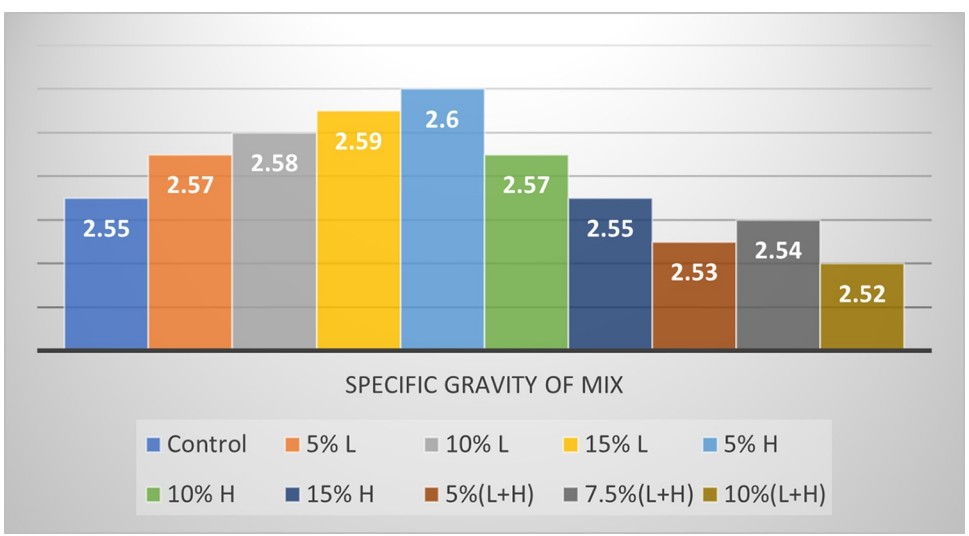

**Fig 2. Specific gravity of the asphalt samples.**

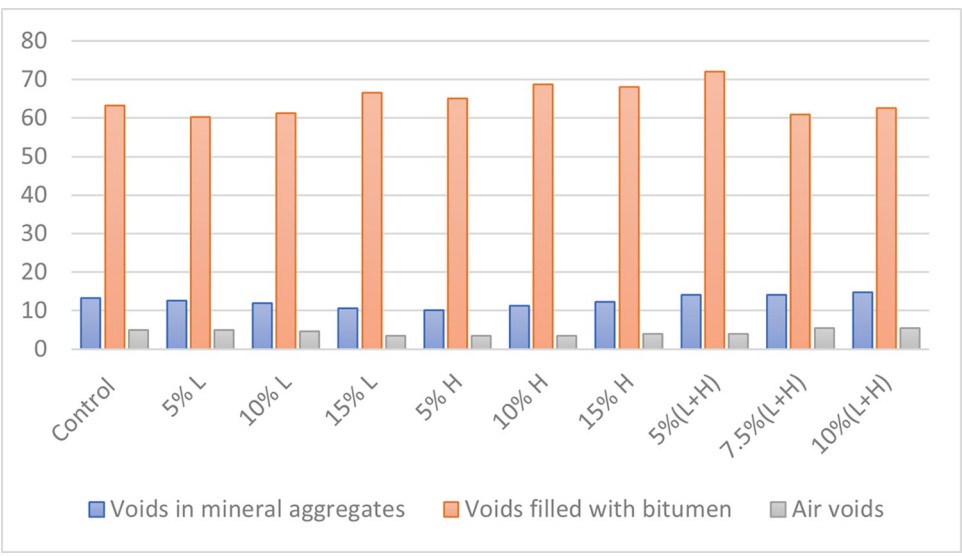

**Fig 3. Void presence in different components of the asphalt samples.**

concrete. The x-ray diffractogram of the control mix and the polymers are given in Figs 5 and 6 which reveals that the crystallographic planes of (020) type did not vary their locations over the series of samples investigated. Fig 5A and 5B shows the XRD patterns of the control sample and the 5 percent RAP + 15 percent HDPE sample respectively. The DSC yielded two peaks at 2 = 29.50 and 45.50, as seen in the diffraction pattern. The strongest diffraction peaks may be seen in the mixture of 5 percent RAP and 15 percent HDPE. Peaks at 29.50 and 45.50 2 d-spacings correspond to crystallographic planes (110) and (020) type and did not move in all of the sample series examined for this material's phase. Because of the HDPE content, their intensity increased. Diffractograms show that the strength of the primary peak varies from sample to sample. Reflections were also seen to broaden, as is usual for smaller and faultier crystalline forms in polymers. Compared to the control mixes, asphalt concrete with 15% HDPE has

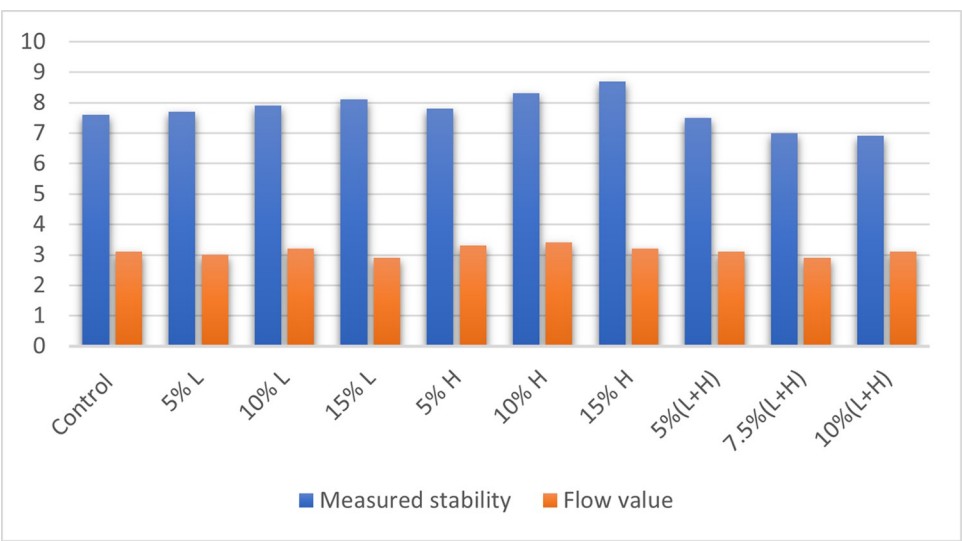

**Fig 4. Stability and flow value of the asphalt samples.**

**Table 4. Results from the Marshall stability test on the asphalt samples.**

| Asphalt design mix | Specific gravity of mix | % Volume of bitumen | % Volume of aggregates | Voids in mineral aggregates | Voids filled with bitumen | Air voids | Measured stability (KN) | Flow value (mm) |
|---|---|---|---|---|---|---|---|---|
| Control | 2.55 | 5.1 | 94.9 | 13.3 | 65.2 | 4.9 | 7.6 | 3.1 |
| 5% L | 2.57 | 5.1 | 94.9 | 12.6 | 60.3 | 5.0 | 7.7 | 3.0 |
| 10% L | 2.58 | 5.1 | 94.9 | 11.9 | 61.3 | 4.6 | 7.9 | 3.2 |
| 15% L | 2.59 | 5.1 | 94.9 | 10.5 | 66.6 | 3.5 | 8.1 | 2.9 |
| 5% H | 2.60 | 5.1 | 94.9 | 10.1 | 65.0 | 3.5 | 7.8 | 3.3 |
| 10% H | 2.57 | 5.1 | 94.9 | 11.2 | 68.7 | 3.5 | 8.3 | 3.4 |
| 15% H | 2.55 | 5.1 | 94.9 | 12.2 | 68 | 3.9 | 8.7 | 3.2 |
| 5%(L+H) | 2.53 | 5.1 | 94.9 | 14 | 72 | 3.9 | 7.5 | 3.1 |
| 7.5%(L+H) | 2.54 | 5.1 | 94.9 | 14.1 | 60.9 | 5.5 | 7.0 | 2.9 |
| 10%(L+H) | 2.52 | 5.1 | 94.9 | 14.7 | 62.5 | 5.5 | 6.9 | 3.1 |

greater high-intensity values. The enhanced stability and rigidity, as well as the reduced flow of HDPE-modified asphalt make it more resistant to permanent deformation. In certain specific applications, asphalt concretes containing HDPE have adequate resistance to permanent deformation and may have better resistance to fracture (reflection cracking and thermal cracking). This finding is consistence with the results of similar studies elsewhere. Fig 6A depicts the X-ray diffraction (XRD) patterns of 5% RAP + 5% LDPE & 5% HDPE. As seen in the diffraction pattern, only one peak at $2\theta = 19.50$ corresponding to the crystallographic planes of (200) was obtained from the DSC. Compared to control mixes, the asphalt concrete's high intensity values were found to be greater, but still lower, than those of HDPE-15 percent. Because of their stronger stability and stiffness and lower flow, HDPE modified asphalt provides another example of their superior capacity to resist permanent deformation. Asphalt concretes containing HDPE are widely thought to be quite beneficial in hot climatic locations for minimising long-term deformations. The diffraction pattern in Fugure 6b shows that the sharp peak obtained from the DSC was around 170 2-Theta. They are close to the values found in similar studies for pure low-density polyethylene. It has been shown that the low intensity values of asphalt concrete with a 15% LDPE mixture shown in Fig 6B are lower than those of control mixes. Low stiffness means LDPE-modified asphalt does not have strong resistance to long-term deformation.

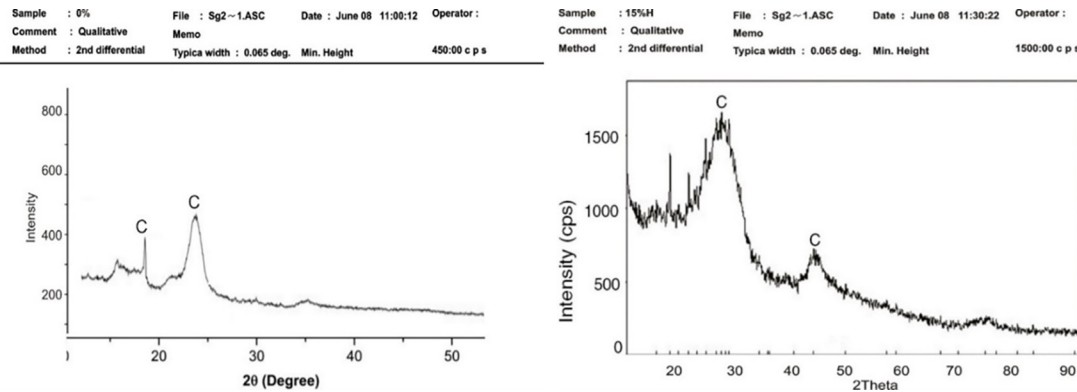

**Fig 5.** X-ray diffraction patterns (a) control mix (b) for 5% RAP + 15% HDPE.

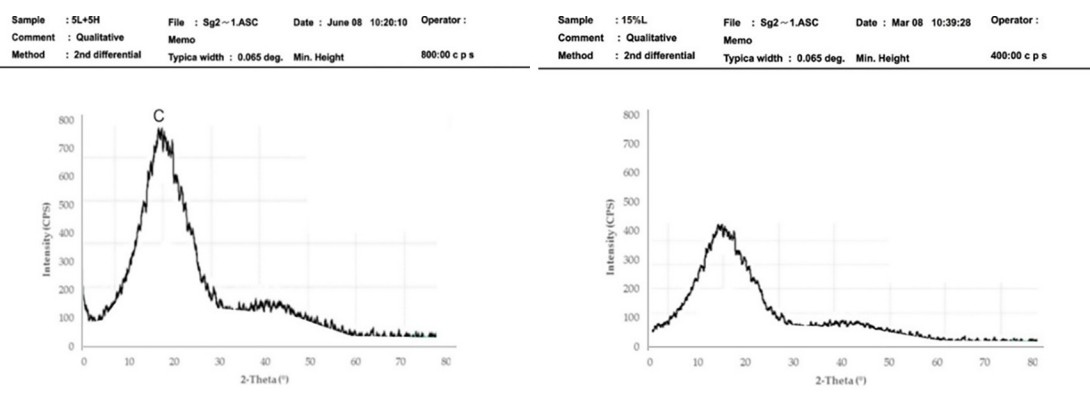

**Fig 6.** X-ray diffraction patterns (a) 5% RAP + 5% LDPE & 5% HDPE (b) 5% RAP + 15% LDPE.

## Surface structure of examination of asphaltic concrete samples using SEM

The structure of an aggregate is an important parameter in determining its suitability for use in asphaltic concrete mix. A Scanning Electron Microscopic (SEM) analysis was performed on all the prepared samples of the asphaltic concrete to determine their microstructure. Fig 7A shows the SEM micrograph of control mix. The structure of the control platelets shows numerous petal-shaped crystals appear to coat the control mix nanoplatelets. The scattered polymer-rich phase was clearly visible in the SEM pictures. Fig 7B shows the SEM micrograph of 5% RAP + 5% LDPE while Fig 7C shows that of 5% RAP + 10% LDPE. HDPE additives were shown to have a worse dispersion in SEM pictures compared to LDPE.

Furthermore, it showed that LDPE concentration of 5% was deemed optimal for greater dispersion. As the surface of modified asphalt appears uneven, SEM scans revealed that LDPE particles were dispersed irregularly in the asphalt binder and had rough surfaces. Other materials were less tightly packed and cohesive with each other. Because of this, the LDPE modified asphalt has uneven particle shape and fragmented porous surface morphology, while HDPE modified asphalt has a smooth surface and reduced segregation. LDPE has a negative influence on engineering qualities such as tensile strength and toughness. It was identified that few particles among the 5% RAP + 10% HDPE (Fig 8C) and 5% RAP + 15% HDPE (Fig 8D) were sporadically larger but more significant in 5% RAP + 10% HDPE. Despite having some coarse particles, this implies they are not significantly affected by the blend. A different shape was totally observed in the SEM micrograph of the control sample (Fig 8A) and the sample with 5% RAP + 5% HDPE (Fig 8B). It was shown that 15% HDPE had superior fracture resistance, more cohesiveness within the binder and a stronger bond to aggregate surfaces than any other binder. This is due to the asphalt being altered to have a greater concentration of HDPE. SEM pictures of the HDPE-modified asphalt revealed a well-distributed and linked nanofiber network in the binder and mixture, contributing to enhanced asphalt concrete adhesion. As a result, as HDPE content rises, so does asphalt adhesion, resulting in improved connection and a more robust network. There are a lot of cone-shaped asphalt structures seen in Fig 8D, which shows that the adhesion bond is equal to the cohesion of the asphalt. The best way to enhance the binder's surface texture was discovered to be a 15 percent HDPE addition. The bigger the HDPE concentration in the binder, the greater the percentage increase in smoothness. An interconnected network will be formed, lowering the length of cracks and increasing asphalt binder's tensile strength. Fig 9A–9D show the SEM micrograph of 5% RAP + 5% LDPE & 5% HDPE (Fig 9B), 5% RAP + 7.5% LDPE & 7.5% HDPE (Fig 9C) and 5% RAP + 10% LDPE & 10% HDPE (Fig 9D). The structure of the control platelets observed in the Fig 9A were quite

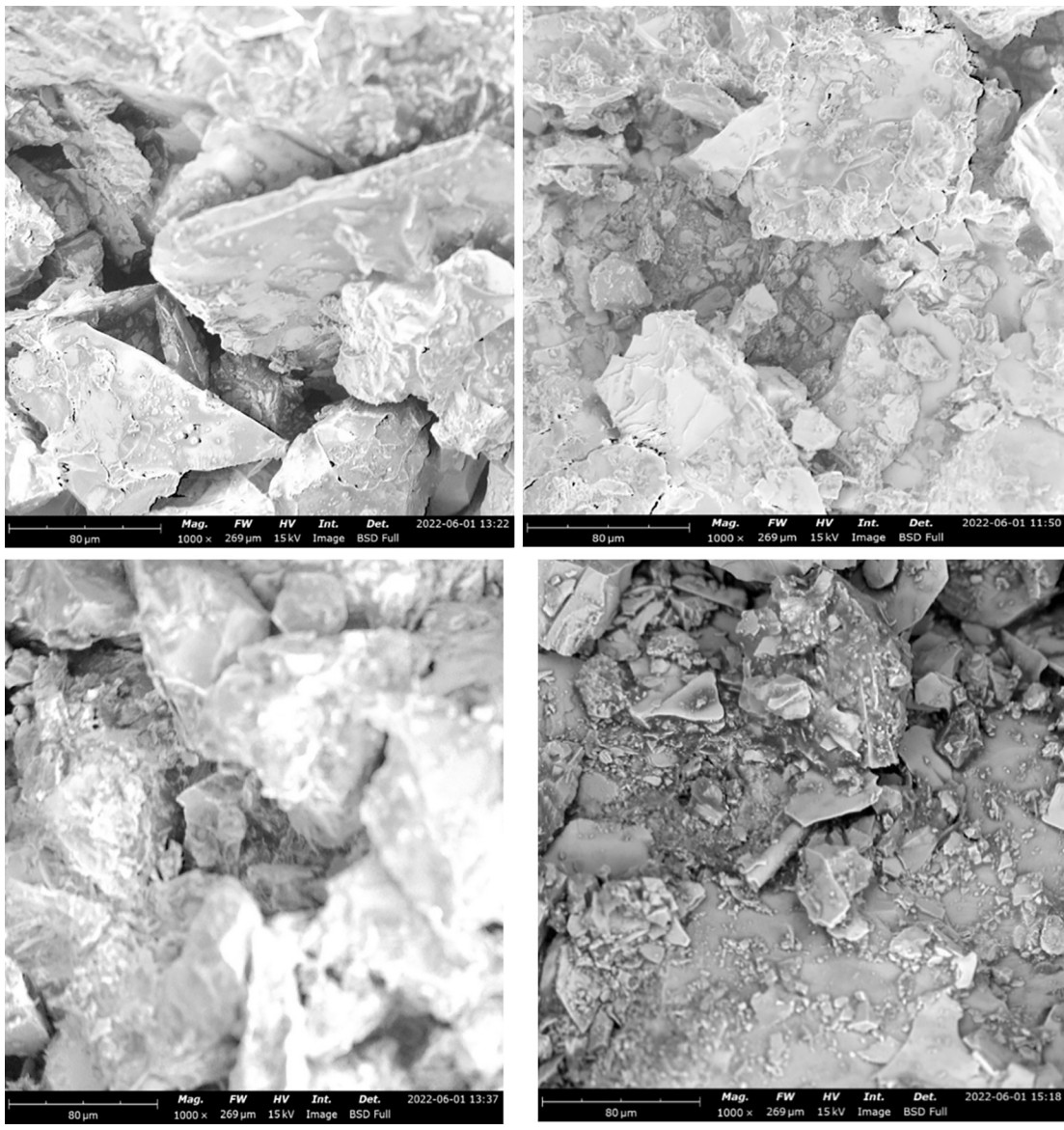

**Fig 7.** SEM image showing different mixtures of RAP and LDPE (a) control mix (b) sample 2—(5% RAP + 5% LDPE) (c) sample 4—(5% RAP + 10% LDPE) (d) sample 6—(5% RAP + 15% LDPE).

similar showing numerous petal-shaped crystals. When compared to SEM photos of asphalt treated with simply HDPE, which were found to be angular with smooth broken surfaces, the LDPE + HDPE images showed a porous and fluffy look. It was seen that there were a few white specks or particles.

## Conclusions

This research work was conducted to determine the feasibility of using recycled asphalt pavement (RAP) and polyethylene polymers (HDPE and LDPE) as partial aggregate replacement in the asphaltic concrete mix. Waste materials such as discarded plastic pipes used in plumbing fixtures served as High-Density Polyethylene (HDPE) while pulverized waste nylon bags served as Low-Density Polyethylene (LDPE). These recovered wastes were utilized in the

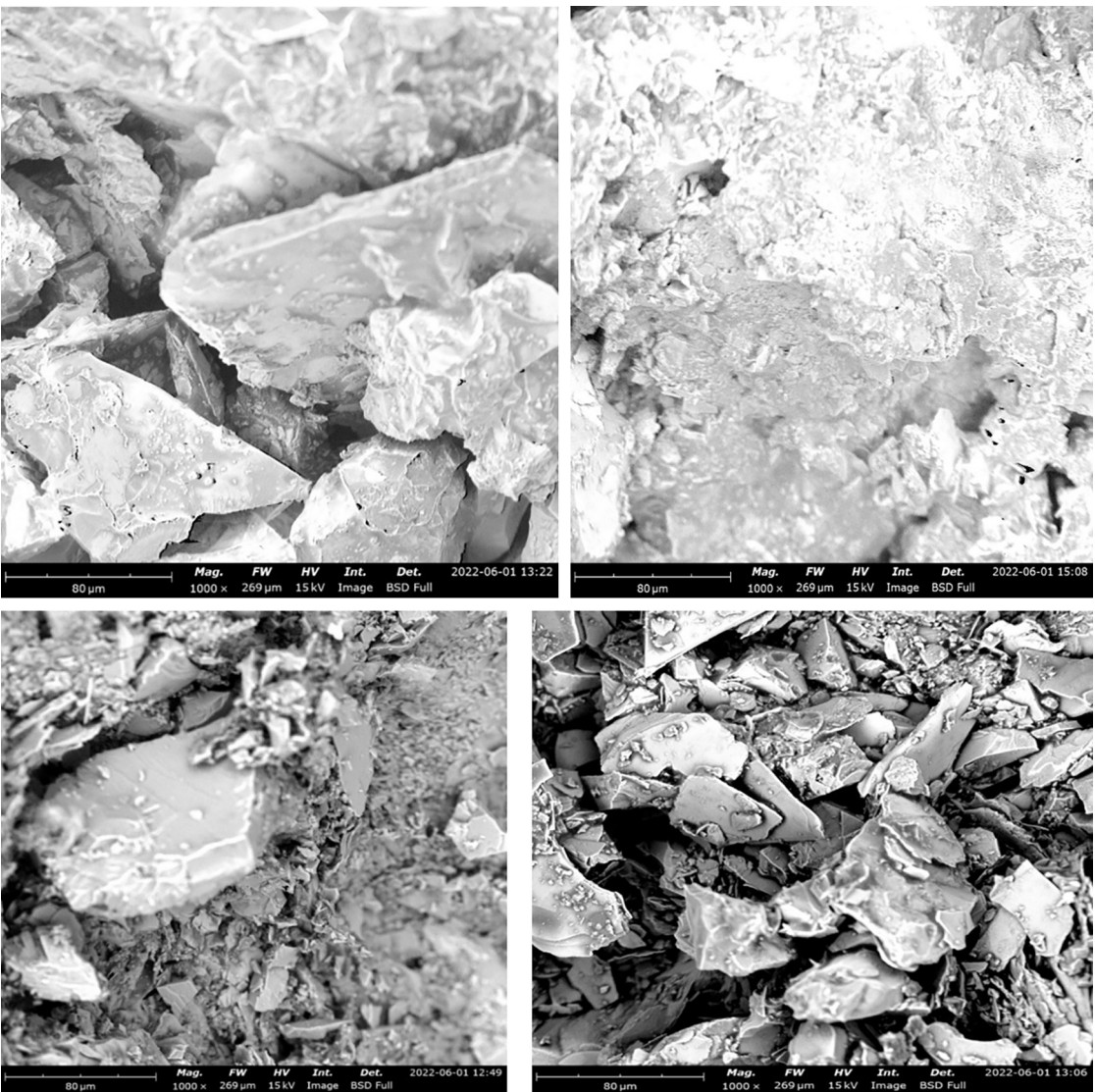

**Fig 8.** SEM image showing different mixtures of RAP and HDPE (a) control mix (b) sample 3—(5% RAP + 5% HDPE) (c) sample 5—(5% RAP + 10% HDPE) (d) sample 7—(5% RAP + 15% HDPE).

asphalt concrete production at different aggregate substitution percentages. The results were then compared to Nigerian general specifications for roads and bridges provided by the federal ministry of works to determine if they meet the acceptable standard for use in road pavement construction. They were also compared to the control sample prepared with no RAP, HDPE, or LDPE substitutes. The following conclusions can be made based on the results from this study:

- The measured stability value of all the asphalt samples passed the minimum specification which is 3.5KN.

- Not all the samples met the required void filled with bitumen range of 65–75, as 5% L, 10% L, 7.5% (L+H) and 10% (L+H) had their voids filled with bitumen below the minimum range which is 65.

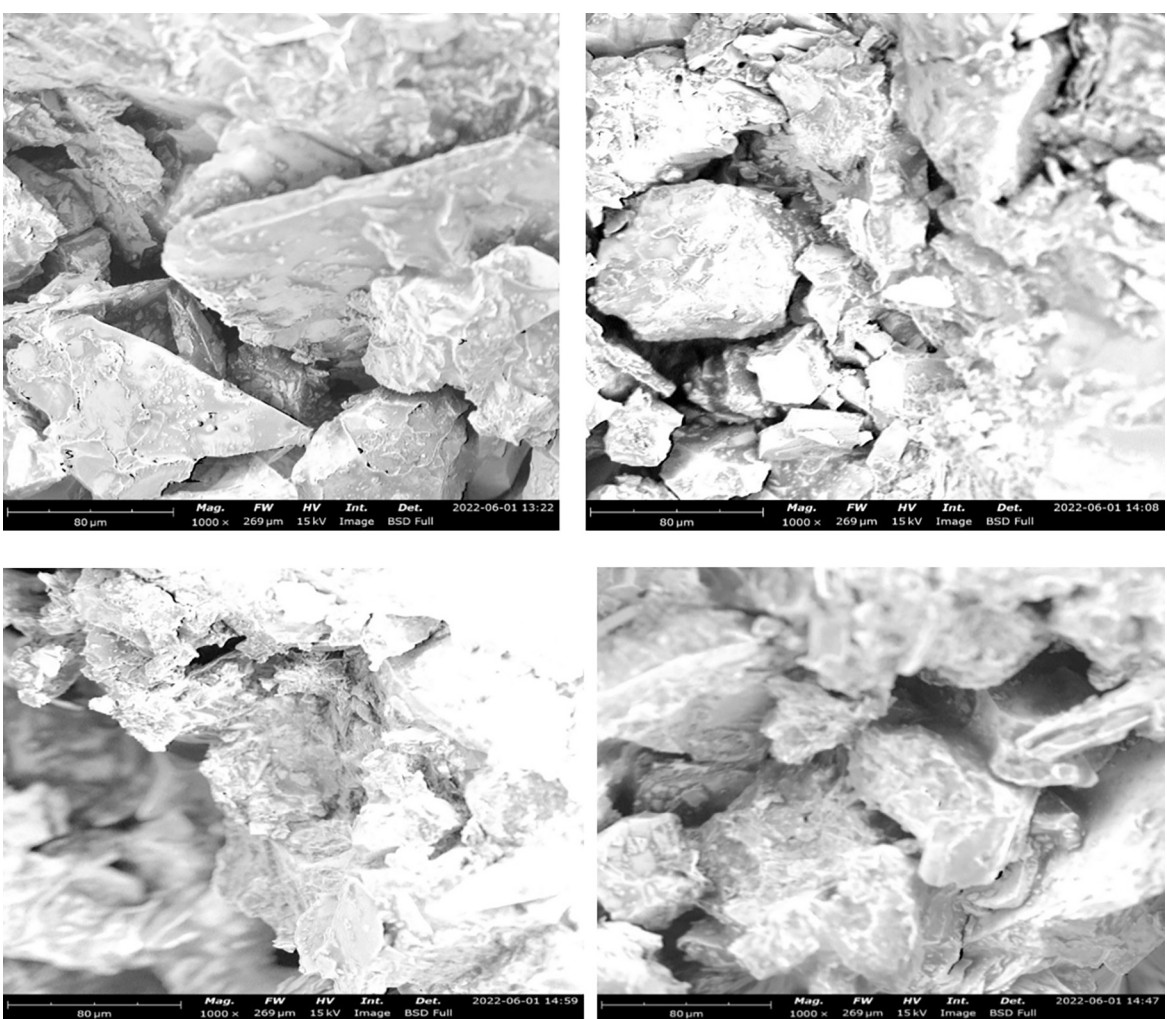

**Fig 9.** SEM image showing different mixtures of RAP, HDPE and LDPE (a) control mix (b) sample 8—(5% RAP + 5% LDPE & 5% HDPE) (c) sample 9—(5% RAP + 7.5% LDPE & 7.5% HDPE) (d) sample 10—(5% RAP + 10% LDPE & 10% HDPE).

- Sample with 15% H which signifies (5% RAP + 15% HDPE) is our optimum asphalt sample, as it had the best stability value of 8.7KN.

- Other important parameters such as the flow, void filled with air, void filled with mineral aggregates, as well as the void filled with bitumen are all within acceptable tolerances.

Therefore, the results obtained from this study demonstrated that RAP, HDPE & LDPE are good substitutes for aggregates in asphaltic concrete production. The asphaltic concrete samples exhibited acceptable material properties and comply with the specified construction material standard. Scaling up and mass adoption of these materials in pavement construction will be of huge environmental benefits in terms of reducing plastic pollution and driving environmental sustainability through materials recovery and reuse in road construction. It is anticipated that future study will assess the long-term degradation of the asphaltic concrete under normal loading conditions.

## Author Contributions

**Conceptualization:** Akanimo Ukpeh.

**Data curation:** Benjamin Oyegbile.

**Formal analysis:** Akanimo Ukpeh.

**Investigation:** Akanimo Ukpeh.

**Project administration:** David Olukanni.

**Resources:** David Olukanni.

**Supervision:** David Olukanni.

**Validation:** Benjamin Oyegbile.

**Writing – original draft:** David Olukanni.

**Writing – review & editing:** Benjamin Oyegbile.

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
