## [Decision Letter · Decision Letter 0]

6 Jul 2023

PONE-D-23-11091Evaluation of Engineering Properties of Asphaltic Concrete Composite Produced from Recycled Asphalt Pavement and Polyethylene PlasticPLOS ONE

Dear Dr. Oyegbile,

Thank you for submitting your manuscript to PLOS ONE. After careful consideration, we feel that it has merit but does not fully meet PLOS ONE’s publication criteria as it currently stands. Therefore, we invite you to submit a revised version of the manuscript that addresses the points raised during the review process.

We look forward to receiving your revised manuscript.

Kind regards,

Anwar Khitab

Academic Editor

PLOS ONE

Journal Requirements:

4. We note that Figure 1 in your submission contain copyrighted images. All PLOS content is published under the Creative Commons Attribution License (CC BY 4.0), which means that the manuscript, images, and Supporting Information files will be freely available online, and any third party is permitted to access, download, copy, distribute, and use these materials in any way, even commercially, with proper attribution. For more information, see our copyright guidelines: http://journals.plos.org/plosone/s/licenses-and-copyright.

Reviewers' comments:

Reviewer's Responses to Questions

**Comments to the Author**

1. Is the manuscript technically sound, and do the data support the conclusions?

Reviewer #1: Yes

Reviewer #2: Yes

2. Has the statistical analysis been performed appropriately and rigorously? 

Reviewer #1: N/A

Reviewer #2: No

3. Have the authors made all data underlying the findings in their manuscript fully available?

Reviewer #1: Yes

Reviewer #2: Yes

4. Is the manuscript presented in an intelligible fashion and written in standard English?

Reviewer #1: Yes

Reviewer #2: Yes

5. Review Comments to the Author

Reviewer #1: 1. add problem statements

2. mentions which layer of pavement can be applied according to the obtained results , in terms of stability and gradation

3. add aggregate gradation according to standard specification.

4. the value of abrasion test is more than 30% ,is there any reasons for that explain.

5. what is the main contribution of this work according to the previous related work.

6. add recommendation

7. explain from your point of view, can we use the RCA in wearing layers for pavement.

8. What about long-term pavement age .. is it suitable or not.

9. I don't see the long term aging test, is there any reason for that.

Reviewer #2: This paper investigates the Evaluation of Engineering Properties of Asphaltic Concrete Composite Produced from Recycled Asphalt Pavement and Polyethylene Plastic. Some comments and questions are listed as following:

1. Correct the line no 74 on page no 3 “Similar communications by Ezemenike et al. and Olukanni et. al (6,7) reported tfindings on the” to “Similar communications by Ezemenike et al. and Olukanni et. al (6,7) reported findings on the”.

2. In the lines 118 to 128 on the page no 5, addition of HDPE and LDPE is described. However, it is recommeneded to describe the mixing techniques with conditions.

3. The aims and objectives of the research articals are limited to components of Marshal stabilty of asphalt mixes and microscopic analysis through SEM and XRD techniques. It would be more reiable to perform few asphalt mixture perofrmance test to investigagete the perofrmance life with addition of modifirers. It is recommended to add atleast one performance test.

4. Physcial properties of bitumen are described in Table 1 on page 5. However, the physical testing after addition of HDPE and LDPE are not added. It is recommended to add such such results for indication of modification.

5. The conclusion sesion requires more deleberated results on the with the scope of study.

6. There is no technique applied for the analysis of results. It is recommneded to add statiscial analysis for better application of results.

7. There are only few references in the introduction section. It is recommneded to add more reference and data relevasnt to scope of the study.

6. PLOS authors have the option to publish the peer review history of their article (what does this mean?). If published, this will include your full peer review and any attached files.

Reviewer #1: No

Reviewer #2: **Yes: **ghulam yaseen

---

## [Author Response · Author response to Decision Letter 0]

27 Sep 2023

The point by point response to reviewers has been attached as a separate PDf file.

---

## [Decision Letter · Decision Letter 1]

27 Oct 2023

Evaluation of Engineering Properties of Asphaltic Concrete Composite Produced from Recycled Asphalt Pavement and Polyethylene Plastic

PONE-D-23-11091R1

Dear Dr. Oyegbile,

We’re pleased to inform you that your manuscript has been judged scientifically suitable for publication and will be formally accepted for publication once it meets all outstanding technical requirements.

Kind regards,

Anwar Khitab

Academic Editor

PLOS ONE

Additional Editor Comments (optional):

Reviewers' comments:

Reviewer's Responses to Questions

**Comments to the Author**

1. If the authors have adequately addressed your comments raised in a previous round of review and you feel that this manuscript is now acceptable for publication, you may indicate that here to bypass the “Comments to the Author” section, enter your conflict of interest statement in the “Confidential to Editor” section, and submit your "Accept" recommendation.

Reviewer #1: All comments have been addressed

Reviewer #2: All comments have been addressed

2. Is the manuscript technically sound, and do the data support the conclusions?

Reviewer #1: Yes

Reviewer #2: Yes

3. Has the statistical analysis been performed appropriately and rigorously? 

Reviewer #1: N/A

Reviewer #2: No

4. Have the authors made all data underlying the findings in their manuscript fully available?

Reviewer #1: Yes

Reviewer #2: Yes

5. Is the manuscript presented in an intelligible fashion and written in standard English?

Reviewer #1: Yes

Reviewer #2: Yes

6. Review Comments to the Author

Reviewer #1: (No Response)

Reviewer #2: The manuscript has been improved. The statistical analysis is still missing. The statistical analysis may me carried out on different samples of Marshal mix with different percentages of modifiers.

7. PLOS authors have the option to publish the peer review history of their article (what does this mean?). If published, this will include your full peer review and any attached files.

Reviewer #1: No

Reviewer #2: No

---

## [Editor Report · Acceptance letter]

6 Feb 2024

PONE-D-23-11091R1 

PLOS ONE

Dear Dr. Oyegbile, 

I'm pleased to inform you that your manuscript has been deemed suitable for publication in PLOS ONE. Congratulations! Your manuscript is now being handed over to our production team.

Kind regards, 

on behalf of

Professor Anwar Khitab 

Academic Editor

PLOS ONE